# Structure-Functional Analysis of Human Cytochrome P450 2C8 Using Directed Evolution

**DOI:** 10.3390/pharmaceutics13091429

**Published:** 2021-09-09

**Authors:** Rowoon Lee, Vitchan Kim, Youngjin Chun, Donghak Kim

**Affiliations:** 1Department of Biological Sciences, Konkuk University, Seoul 05029, Korea; pharro96@naver.com (R.L.); kbc0829@konkuk.ac.kr (V.K.); 2College of Pharmacy, Chung-Ang University, Seoul 06974, Korea; yjchun@cau.ac.kr

**Keywords:** P450, CYP2C8, directed evolution, luciferin, paclitaxel, arachidonic acid, mass spectrometry

## Abstract

The human genome includes four cytochrome P450 2C subfamily enzymes, and CYP2C8 has generated research interest because it is subject to drug–drug interactions and various polymorphic outcomes. To address the structure-functional complexity of CYP2C8, its catalytic activity was studied using a directed evolution analysis. Consecutive rounds of random mutagenesis and screening using 6-methoxy-luciferin produced two mutants, which displayed highly increased luciferase activity. Wild-type and selected mutants were expressed on a large scale and purified. The expression levels of the D349Y and D349Y/V237A mutants were ~310 and 460 nmol per liter of culture, respectively. The steady-state kinetic analysis of paclitaxel 6α-hydroxylation showed that the mutants exhibited a 5–7-fold increase in *k*_cat_ values and a 3–5-fold increase in catalytic efficiencies (*k*_cat_/*K*_M_). In arachidonic acid epoxidation, two mutants exhibited a 30–150-fold increase in *k*_cat_ values and a 40–110-fold increase in catalytic efficiencies. The binding titration analyses of paclitaxel and arachidonic acid showed that the V237A mutation had a lower *K*_d_ value, indicating a tighter substrate-binding affinity. The structural analysis of CYP2C8 indicated that the D349Y mutation was close enough to the putative binding domain of the redox partner; the increase in catalytic activity could be partially attributed to the enhancement of the P450 coupling efficiency or electron transfer.

## 1. Introduction

Cytochrome P450 enzymes (P450, CYP) consist of a superfamily of heme-thiolate proteins and exhibit a unique spectral absorbance at 450 nm when their reduced forms are combined with carbon monoxide [1,2]. The human genome contains 57 P450 genes, which are distributed in most autosomal chromosomes [3]. Among the many P450s, 12 enzymes from the P450 1, 2, and 3 families play central roles in the oxidative metabolism of xenobiotic chemicals, including most clinical drugs [4].

There are four genes of the cytochrome P450 2C subfamily of enzymes in the human genome, which are located in chromosome 10q24 in the centromere-2C18-2C19-2C9-2C8-telomere order [5]. Despite the considerable similarity of sequences among the P450 2C enzymes, each P450 2C enzyme has a unique substrate specificity and role in drug metabolism [6]. Cytochrome P450 2C8 (CYP2C8) accounts for approximately 6–7% of the total P450 content expressed in the human liver [7]. It is known to catalyze the metabolism of more than 100 drugs, including paclitaxel, amodiaquine, cerivastatin, enzalutamide, and pioglitazone. The oxidation reaction of paclitaxel, an anticancer agent that produces 6α-hydroxypaclitaxel, is a typical biomarker reaction of CYP2C8 [8]. CYP2C8 also participates in the conversion of arachidonic acid to biologically active epoxyeicosatrienoic acids (EETs), which are involved in the regulation of physiological processes, including blood pressure regulation, platelet aggregation, vascular function, and pancreatic peptide hormone secretion [9,10,11]. To date, 18 allelic variants of CYP2C8 have been identified, and these allelic variants have shown potential substrate dependencies and various clinical outcomes (https://www.pharmvar.org/gene/CYP2C8) (accessed date: 1 August 2021).

In the structure-function analysis, the conformation of the peptide backbone of CYP2C8 is very similar to that of CYP2C19, but its active site is much larger than those of CYP2C9 and CYP2C19 [12]. The active site of CYP2C8 is about 1400 Å^3^ in size, more rigid, and L-shaped [4]. This difference is mainly due to alternations in the amino acid residues that form the substrate-binding cavities in CYP2C8. The functional role of Arg241 in substrate specificity has been previously noted [13]. In addition, two molecules of fatty acids are bound in the dimmer interface in the crystal structure of CYP2C8, which has also been observed in solution [13]. These studies of the structure-function in CYP2C8 have contributed to understanding the substrate specificities and drug–drug interactions in P450 metabolism.

Directed evolution is a widely used method in the field of protein engineering as well as in the analysis of structure–function relationships [14]. This approach has been successfully adopted to determine any alterations in the functions of enzymes such as substrate specificity, catalytic activity, thermal stability, enantioselectivity, and pH profile [15]. It consists of sequential cycles of random mutagenesis and screening. The first step is the construction of a mutant library by performing random mutagenesis with an error-prone polymerase chain reaction (PCR) [16]. The screening and selection of the library are then performed to identify the desired mutations. The most important requirement for a successful directed evolution experiment is the selection of appropriate and efficient screening methods for the enzymatic function of interest.

In this study, directed evolution consisting of random mutagenesis and luminescence-based high-throughput screening was applied to human CYP2C8 to study the structure–function relationships in P450 enzymes. CYP2C8 mutants with enhanced catalytic activity were selected. Through a DNA sequence analysis and an X-ray crystal structure comparison, the positions of the mutated amino acids were identified, and their functions were speculated.

## 2. Materials and Methods

### 2.1. Chemicals

Arachidonic acid, 1,2-dilauroyl-sn-glycero-3-phosphocholine (DLPC), and d-glucose 6-phosphate were purchased from Sigma-Aldrich (St. Louis, MO, USA). We also purchased 3-[(3-Cholamidopropyl) dimethylammonio]-1-propanesulfonate (CHAPS) from GoldBio (St. Louis, MO, USA), while paclitaxel and dibenzylfluorescein (DBF) were purchased from Santa Cruz Biotechnology (Dallas, TX, USA). Luciferin-ME and luciferin detection reagent (LDR) were purchased from Promega (Madison, WI, USA). The primary vascular eicosanoid LC–MS mixture and (±)11, (12)-epoxy-5Z,8Z,14Z-eicosatrienoic acid (11,12-EET) were purchased from Cayman Chemicals (Ann Arbor, MI, USA). Professor Sang Kyum Kim of Chungnam National University (Daejon, Korea) kindly provided 6α-Hydroxypaclitaxel. Ni^2+^-nitrilotriacetate (NTA) agarose was purchased from Thermo Fisher Scientific (Waltham, MA, USA). Rat NADPH-P450 reductase (NPR) was heterologously expressed in *Escherichia coli* HMS174 (DE3) and purified, as described previously [17]. In brief, the *E. coli* HMS174 (DE3) strains transformed with the pOR263/NPR plasmid were inoculated into terrific broth (TB) media containing 50 μg/mL ampicillin (Sigma-Aldrich, St. Louis, MO, USA), 1 μg/mL riboflavin (Sigma-Aldrich), and 200 mg/mL glucose (Sigma-Aldrich); these were grown at 37 °C, with shaking at 250 rpm until reaching an OD_600_ ranging between 0.6 and 0.8, before inducing the expression by adding 1 mM of the IPTG solution (Sigma-Aldrich), and reducing the temperature to 28 °C. The culture was subsequently incubated overnight, with shaking at 200 rpm, and the expression was carried out at 28 °C for 20 h before harvesting. The bacterial inner membrane fractions containing NPR were isolated and prepared by ultracentrifugation (10^5^× *g*, 90 min). The prepared membrane fractions were solubilized at 4 °C overnight, in a 100 mM potassium phosphate buffer (pH 7.4) containing 20% (*w*/*v*) glycerol (Sigma-Aldrich), 0.5 M NaCl (Sigma-Aldrich), 10 mM β-mercaptoethanol (Sigma-Aldrich), 1.0% (*w*/*v*) CHAPS (Anatrace, Maumee, OH, USA), and 0.5% (*v*/*v*) Tergitol NP-10 (Sigma-Aldrich). The purification of the NPR enzyme using an ADP-agarose column chromatography was carried out.

### 2.2. Construction of P450 2C8 Random Mutant Libraries

The open reading frame region of CYP2C8 (1.5 kb) was randomly mutated and amplified using Mutazyme™ II DNA polymerase (Agilent Technologies, Santa Clara, CA, USA) according to the manufacturer’s instructions, with the forward primer 5′-GCGAGGTCATATGGCTCTG-3′ and the reverse primer 5′-CCCTGGTTCTAGACTAATGG-3′. The amplified PCR products were purified and cloned into the pCW bicistronic vector using the *Nde*I and *Xba*I restriction enzyme sites. The ligated library plasmid was transformed into *E. coli* DH10b ultracompetent cells (Life Technologies, Carlsbad, CA, USA) and subsequently amplified.

### 2.3. Screening of CYP2C8 Libraries

The constructed random mutant libraries of CYP2C8 were transformed into *E. coli* DH5α cells. The colonies on agar plates with LB media were inoculated into each well of the 96-well tissue culture test plates filled with 200 μL of TB media, including ampicillin (50 μg/mL), 0.5 mM 5-aminolevulinic acid (5-ALA, Sigma-Aldrich), 1.0 mM isopropyl β-d-thiogalactoside (IPTG), 1.0 mM thiamine, and trace elements. Approximately 300 colonies were picked for each round of screening. Randomly mutated CYP2C8 enzymes were expressed at 28 °C for 48 h, and then the culture media were removed by centrifugation. The luminescent activity of the mutants was measured by adding a 2× reaction mixture (M9 minimal medium (including 1% glucose, *w*/*v*) and luciferin-ME (300 μM)) to each well of the 96-well plates. The reaction was stopped by adding an LDR (including luciferase, P450-GloTM Assays) after incubation at 37 °C for 4 h, and luminescence was detected using a Multi-Mode Microplate Reader (BioTek, Winooski, VT, USA).

### 2.4. Expression and Purification of Wild-Type and Mutant CYP2C8

*E. coli* DH5α strains transformed with plasmids carrying the cDNA of wild-type and selected mutant CYP2C8s were inoculated into LB media containing 50 μg/mL ampicillin and incubated overnight at 37 °C. After an overnight incubation, the cultures were transferred into 500 mL of TB containing 50 μg/mL ampicillin. The expression cultures were incubated at 37 °C and 230 rpm, and the supplements containing 1.0 mM IPTG, 0.5 mM 5-ALA, 1.0 mM thiamine, and trace elements were added to induce the expression of P450 enzymes. The expression cultures were further grown at 28 °C and 200 rpm for 24 h; then, the bacterial inner membrane fractions containing wild-type and mutant CYP2C8s were obtained using a previously described method [18]. The prepared bacterial membrane fractions were solubilized overnight at 4 °C in the 100 mM potassium phosphate buffer (pH 7.4) containing 0.1 mM ethylenediaminetetraacetic acid, 10 mM β-mercaptoethanol, 1.5% CHAPS (*w*/*v*), and 20% glycerol. The wild-type and mutant enzymes of CYP2C8 were purified using an Ni^2+^-NTA column according to the method described previously [19,20]. In brief, the supernatant from ultracentrifugation, including the solubilized proteins, was loaded onto a Ni^2+^-NTA agarose column that was pre-equilibrated with the 100 mM potassium phosphate buffer (pH 7.4) containing 5 mM imidazole (Sigma-Aldrich). Once samples were loaded onto the column, they were washed with the 100 mM potassium phosphate buffer (pH 7.4) containing 20 mM imidazole, and then eluted with 300 mM imidazole. The eluted protein was dialyzed in the 100 mM potassium phosphate buffer (pH 7.4) containing 20% glycerol at 4 °C.

### 2.5. Catalytic Activity Assay

Catalytic activities were determined using purified CYP2C8 enzymes and rat NPR in phospholipid reconstituted systems [21]. For luciferin-ME *O*-demethylation, the reaction mixtures included 10 pmol of purified 2C8 enzyme, 20 pmol of rat NPR, and 1,2-dilauroyl-*sn*-glycero-3-phosphocholine (DLPC, 100 μg/mL), together with 50 μM luciferin-ME in 50 μL of the 100 mM potassium phosphate buffer (pH 7.4). The reactions were initiated by adding the NADPH-generating system (100 mM glucose 6-phosphate, 10 mM NADP^+^ (Sigma-Aldrich), and 1 mg/mL glucose-6-phosphate dehydrogenase (Sigma-Aldrich) after pre-incubation at 37 °C for 3 min. The reaction was carried out at 37 °C for 10 min. Then, 50 μL of LDR was added and a luminescent reaction was initiated, which was read using a Multi-Mode Microplate Reader (BioTek, Winooski, VT, USA).

The catalytic activities of paclitaxel 6α-hydroxylation were analyzed using ultra-performance liquid chromatography–tandem mass spectrometry (UPLC–MS/MS) (Waters, Milford, MA, USA). The reaction mixtures consisted of 50 pmol purified CYP2C8, 100 pmol purified rat NPR, and DLPC, with various concentrations of paclitaxel in 500 μL of the 100 mM potassium phosphate buffer (pH 7.4). The reactions were initiated by adding the NADPH-generating system after pre-incubation at 37 °C for 3 min. The reaction was incubated for 5 min at 37 °C and terminated by adding 1 mL of CH_2_Cl_2_. After vortexing, the samples were centrifuged at 3500 rpm for 15 min. The organic phase of each sample was transferred to a new test tube and dried under N_2_ gas. The dried products were dissolved in 200 μL CH_3_CN. These samples were then injected into an ACQUITY UPLC™ BEH C18 column (50 × 2.1 mm, 1.7 μm) equipped with Waters ACQUITY UPLC™ (Waters, Milford, MA) and Waters Quattro Premier™ (Waters). The mobile phase consisted of H_2_O containing 10% (*v*/*v*) CH_3_CN (with 0.1% formic acid) (A) and 100% CH_3_CN (with 0.1% formic acid) (B), at a flow rate of 0.3 mL/min. Mobile phase A was held at 50% during the first 0.5 min and decreased to 30% for 4 min. The analytes were observed using positive electrospray ionization and multiple reaction mode. The source temperature was 120 °C, the desolvation temperature was 380 °C, the desolvation gas flow rate was 550 L/h, and the cone gas flow rate was 50 L/h. The column temperature was maintained at 40 °C. The positive ionization transitions of paclitaxel (*m*/*z* 854.4 > 286) and 6α-hydroxypaclitaxel (*m*/*z* 870 > 286) were monitored at collision energies of 20 and 4 eV, respectively. The peak areas were calculated using the QuanLynx software (Waters).

For arachidonic acid epoxidation, the reaction mixtures consisted of 50 pmol purified CYP2C8, 100 pmol purified rat NPR, and DLPC, with various concentrations of arachidonic acid in 500 μL of the 100 mM potassium phosphate buffer (pH 7.4). After pre-incubation at 37 °C for 3 min, the reactions were initiated by adding the NADPH-generating system. The enzymatic reaction was carried out at 37 °C for 10 min and was terminated by adding 1 mL of CH_2_Cl_2_. After vortexing, the samples were centrifuged at 3500 rpm for 15 min. The organic phase of each sample was transferred to a new test tube and dried under N_2_ gas. The dried products were resuspended in 200 μL methanol and analyzed using UPLC–MS/MS. The negative ionization transitions of arachidonic acid (*m*/*z* 303.3 > 259.3); 11, 12-EET (*m*/*z* 319.3 > 167.3); and 14, 15-EET (*m*/*z* 319.3 > 175.3) were monitored at collision energies of 15, 13, and 13 eV, respectively.

### 2.6. Binding Spectral Titration of Wild-Type and Mutant CYP2C8s

Binding titration analysis was performed using purified wild-type and mutant CYP2C8s. CYP2C8 enzymes were diluted to 5 μM in the 100 mM potassium phosphate buffer (pH 7.4) and then dispensed into two glass cuvettes. The spectroscopic changes from 350 to 500 nm were recorded using a Cary 100 spectrophotometer (Varian, Inc., Palo Alto, CA, USA), with subsequent additions of paclitaxel or arachidonic acid. The differences in the absorbance between the maximum and minimum wavelengths were plotted versus the substrate concentration to estimate the substrate binding affinities (*K*_d_).

### 2.7. Molecular Docking Modeling of CYP2C8

Docking analysis of the CYP2C8 D349Y/V237A mutant with paclitaxel and arachidonic acid were performed with the Autodock 4.2 software (The Scripps Research Institute, La Jolla, CA, USA). The structural coordinate of the V237A/D349Y mutant was obtained from CYP2C8 (pdb ID 1PQ2) using the Coot program [22]. Paclitaxel and the arachidonic acid molecule were taken from the PDB (Protein data bank, pdb IDs are 3J6G and 5ECF) and used for docking. Before docking, all water molecules were removed from the PDB, except for the prosthetic heme group.

## 3. Results

### 3.1. Random Mutagenesis and Selection of CYP2C8 Mutants

Luciferin-ME was converted to luciferin by CYP2C8, and luminescence emission was achieved using luciferase contained in the LDR (Figure 1A). As a result of random mutagenesis in the open reading frame of the wild-type CYP2C8, the first mutant library was generated. After expressing mutants of the CYP2C8 proteins in *E. coli* DH5α, the luminescence generated by the reaction with luciferin-ME was measured at the whole-cell level (Figure 1A). A mutant was selected in the first round of screening, and it yielded an increased activity approximately 10-fold higher than that of the wild-type CYP2C8 (Figure 1B). The result of the DNA sequence analysis of the selected mutant showed that the Asp349 of CYP2C8 was replaced by Tyr. An error-prone PCR was performed using the open reading frame of the CYP2C8 mutant—D349Y. The second mutant library was generated, and screening was carried out in the same manner. A mutant clone with an activity higher than that of the D349Y was selected. The DNA sequence analysis showed that the mutant accumulated mutations in the following amino acid residue of CYP2C8: D349Y/V237A. In the reaction with luciferin-ME at the whole-cell level, the CYP2C8 mutant—D349Y/V237A—showed activity approximately 18 times higher than that of the D349Y (Figure 1B).

### 3.2. Expression and Purification of Recombinant Wild-Type and Mutant CYP2C8s

Recombinant wild-type enzymes and the selected CYP2C8 mutants were expressed in *E. coli* cells. The expression levels of the D349Y and D349Y/V237A mutants were ~310 and 460 nmol/L, respectively, while that of the wild type was ~500 nmol/L (Figure 2). Purified wild-type and selected mutant CYP2C8s were obtained using Ni^2+^-NTA affinity column chromatography, and CO-binding spectra exhibited a distinct Soret peak at 450 nm; however, no 420 peak was observed (data not shown).

### 3.3. Catalytic Activities of Selected CYP2C8 Mutants

The catalytic activities of CYP2C8 were analyzed using purified P450 enzymes from the selected mutants. The demethylation activity of luciferin-ME was measured after normalizing the P450 concentration (10 pmol) of the CYP2C8 mutants. Purified D349Y and D349Y/V237A mutant enzymes also showed highly enhanced activities with 19-fold and 190-fold increases, respectively (Figure 3A).

Paclitaxel is a typical biomarker of CYP2C8 metabolism. Paclitaxel 6α-hydroxylation was analyzed using a reconstitution system. In the chromatogram of UPLC-MS/MS, the retention times of 6α-hydroxypaclitaxel and paclitaxel were 1.48 and 1.95 min, respectively (data not shown). Steady-state kinetic analysis of paclitaxel 6α-hydroxylation by wild-type and mutant CYP2C8s indicated that the catalytic efficiencies (*k*_cat_/*K*_M_) of the D349Y mutant showed a four-fold increase compared with the wild type, mainly due to an increase in the turnover number (Figure 3B and Table 1). However, D349Y/V237A did not enhance the activity of D349Y, the template clone, for paclitaxel 6α-hydroxylation (Figure 3B and Table 1).

CYP2C8 preferentially epoxidizes arachidonic acid as an endogenous substrate and produces EETs, mainly 11,12-EET (ω-9 epoxidation) and 14,15-EET (ω-6 epoxidation). The epoxidation activities of CYP2C8 were analyzed using UPLC-MS/MS. A steady-state kinetic analysis of arachidonic acid epoxidation showed a high increase in activities in both the D349Y and D349Y/V237A mutants (Figure 3C,D and Table 1). When comparing the catalytic efficiencies (*k*_cat_/*K*_M_), the D349Y mutant showed an a 67-fold increase in 11,12-EET and a 40-fold increase in 14,15-EET, respectively. In the case of the D349Y/V237A mutant, the turnover numbers (*k*_cat_) were also enhanced (3–4 times), but the *K*_M_ values of D349Y/V237A also increased; therefore, only a slight alteration was observed in the catalytic efficiency compared with the D349Y mutant (Figure 3C,D and Table 1).

### 3.4. Binding of Substrate to CYP2C8 Mutants

The binding affinities of purified wild-type and selected mutant CYP2C8s to paclitaxel and arachidonic acid were determined (Figure 4 and Table 2). The binding titrations of paclitaxel and arachidonic acid to CYP2C8 showed type I spectral changes, with an increase at ~380 nm and a decrease at ~425 nm (Figure 4). D349Y showed similar binding affinities to those of the wild type, both in paclitaxel and arachidonic acid (Table 2). However, the D349Y/V237A mutant exhibited much lower *K*_d_ values, indicating a tighter substrate-binding affinity (Table 2). This result suggests that the V237A mutation has an effect on substrate binding.

## 4. Discussion

In this study, the structure–function relationship of the human CYP2C8 was studied using a luminescence-based directed-evolution analysis. A mutant with Asp 349 substituted by Tyr was identified in the first-generation screening, and an additional mutation of Val 237Ala was subsequently identified in the second-generation screening. These mutations exhibited significant effects on the catalytic activities found in the metabolic reactions of typical xenobiotic and endogenous substrates of CYP2C8.

The X-ray crystal analysis of CYP2C8 illustrated the binding structures of CYP2C8 complexed with its substrates or inhibitor 9-*cis*-retinoic acid, felodipine, troglitazone, and montelukast in a large, active-site cavity [12,13,23]. The V237A mutation is located in helix G of CYP2C8 (Figure 5, PDB ID: 2NNH). The F–G region is one of the structures that form the outer boundary of the substrate-binding cavity and also the channel architecture of P450 enzymes, which varies between mammalian P450s as a result of the plasticity of P450 [24]. Johnson et al. reported that the flexibility of the helices of the F–G region is relatively low in the X-ray crystal structure of CYP2C8 because the region near helices F–G is a dimer-forming domain with other CYP2C8 molecules (Appendix A) [23]. One postulation of the V237A mutation is that the replacement of a smaller hydrophobic residue (alanine) by valine may endow the helix G region with more flexibility, which may cause an altered interaction with the substrate. In order to understand the effect of the V237A mutation, molecular docking models were constructed using the reported X-ray crystal structure of CYP2C8 (pdb ID 1PQ2). The best docking models were selected on the basis of the substrates being oriented towards the heme for the productive reaction with the lowest RMSD and minimum energy. In the docking models, paclitaxel interacted with the Ala237 residue and heme, with close distances of 4.4 Å and 4.5 Å, respectively (Figure 5A). The docking of arachidonic acid also showed the similar distances of 4.3 Å to the Ala237 residue (Figure 5B). These docking models imply that the V237A mutation can modulate substrate binding in the active site access channel. For this reason, the effect of the V237A mutation produced tighter binding affinities with paclitaxel and arachidonic acid (Table 2). However, tighter binding resulted in an increase in the turnover number only in arachidonic acid epoxidation, but not in paclitaxel 6α-hydroxylation (Figure 3 and Table 1).

The D349Y mutation is located in helix K of CYP2C8 (Appendix A). The helix K of the human P450 enzyme is involved in the binding with redox partner proteins [25,26]. In the structure of human CYP17A1, the positively charged residues Arg347 and Arg358 are located in helix K, and they play an important role in interacting with NADPH-P450 reductase and cytochrome *b*_5_ as a basic “patch” structure [25,26]. The Asp349 residue in CYP2C8 is located very close to this basic patch; presumably, it may help in the precise control of the interaction with reductase partners. Tyrosine is a neutral aromatic amino acid with a large size; therefore, it is possible for the surface charge adjacent to Asp349 to be efficiently masked. Based on the structural analysis of the D349Y mutation, the speculation on improved catalytic activity is that this mutation might contribute to the enhancement of electron transfer or electron coupling efficiency [27]. However, the results of stop-flow and coupling-efficiency experiments need to be analyzed in future studies.

The expression levels of the selected CYP2C8 mutants in *E. coli* cells were similar to those of the wild type (Figure 2). Random mutagenesis frequently affects the structural folding of proteins to produce unstable or apoenzyme species implicated by a 420 peak [28]. As indicated by the structural model, two mutated residues in D349Y/V237A in the outer locations of the protein did not perturb the structural stability of P450 in the process of improving the catalytic activity. Our previous study on CYP2A6 showed that the enhancement of turnover numbers for improved enzymatic performance was achieved by sacrificing the tight binding affinity of the substrate [29]. However, the current study suggests that the catalytic evolution of the P450 enzyme is possible without affecting the substrate-binding affinity (Figure 4 and Table 2).

Successful directed evolution studies always require efficient high-throughput screening, and finding a suitable screening method proved to be the most challenging aspect [30]. Previously, we applied the genotoxicity and fluorescence induced by particular P450 activities [21,28]. Finding the genotoxic or fluorescent substrate for the CYP2C8 activity failed. However, the enzymatic activities of the CYP2C8 mutants selected by a luminescence substrate demonstrated that 6-methoxy-luciferin was highly correlated with clinical or endogenous substrates of CYP2C8 (Figure 3). Efforts on establishing structure–function relationships in P450s with directed evolution often use structural models, as shown in this study. We believe that the prediction of mutated locations in the CYP2C8 structure can give an insight into understanding how the mutated residues exercise their role in the alternation of enzyme activities. Therefore, the use of directed evolution combined with structural information could give a more detailed explanation of the roles of amino acids in protein.

Recently, many artificial intelligence techniques have been developed, and they have been widely applied to various relationship analyses and used for the repositioning of FDA-approved drugs [31,32,33]. Therefore, cooperative work with artificial intelligence, particularly in the initial screening for the relationship and in the analysis of the structure–function relationship, should be pursued to elaborate the directed evolution approach to CYP2C8 in the future.

## 5. Conclusions

In conclusion, a directed evolution analysis based on random mutagenesis and luminescence assay-based high-throughput screening was applied to study the structure–function relationship of CYP2C8. Two mutations on the outer surface of CYP2C8 (not on the active site) were identified, all of which affected the catalytic activity of CYP2C8. The results of this study indicate that the modulation of architectures in the substrate access channel and the reduction partner binding domain can dramatically influence the catalytic activities, but the outcomes of the altered activities are dependent on the structural diversity of various substrates.

## Figures and Tables

**Figure 1 pharmaceutics-13-01429-f001:**
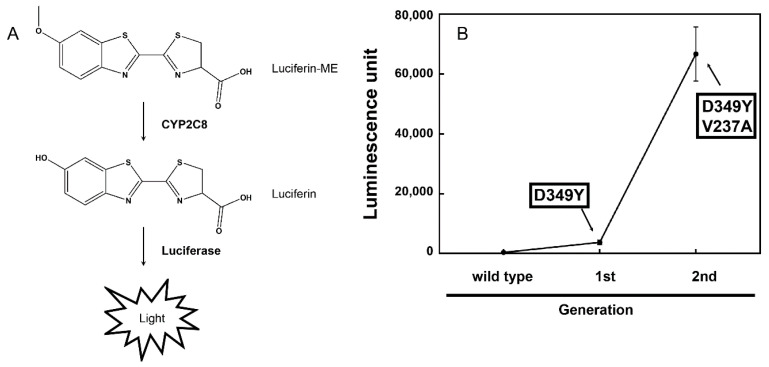
Directed evolution of CYP2C8. (**A**) Enzymatic reaction of the luciferin-ME O-demethylation by CYP2C8. Luciferase in the luciferin detection reagent (LDR) emits light via luciferin produced by CYP2C8. (**B**) Process of directed evolution of CYP2C8 random mutants for the luciferin-ME O-demethylation activity. After each round of screening, the luminescent activity of the mutants was remeasured in triplicate assays.

**Figure 2 pharmaceutics-13-01429-f002:**
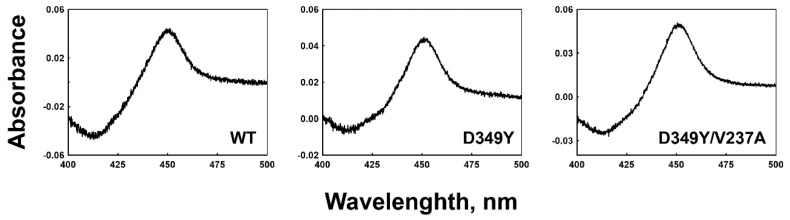
CO-binding spectra of wild-type and selected mutant CYP2C8s in the *E. coli* whole-cell level. Expression levels of the wild type, D349Y, and D349Y/V237A were ~500, 310, and 460 nmol /liter of culture, respectively.

**Figure 3 pharmaceutics-13-01429-f003:**
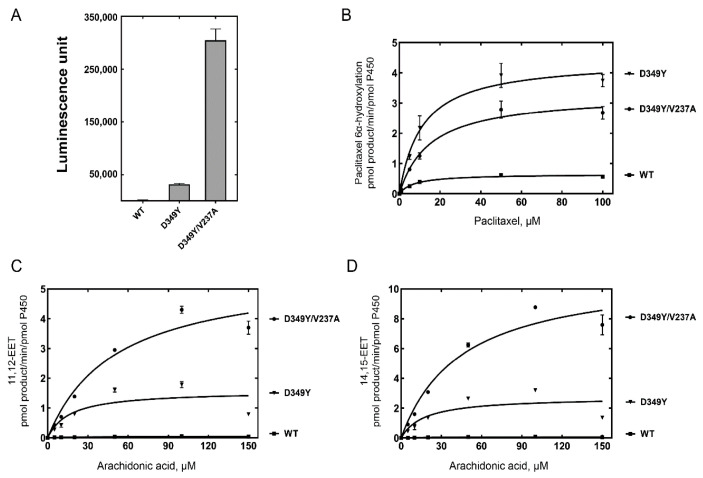
Catalytic activities of purified CYP2C8 mutant enzymes. (**A**) Luciferin-ME *O*-demethylation by purified CYP2C8 mutant enzymes. (**B**) Steady-state kinetic analysis of paclitaxel 6α-hydroxylation. (**C**) Steady-state kinetic analysis of arachidonic acid ω-9 epoxidation. (**D**) Steady-state kinetic analysis of arachidonic acid ω-6 epoxidation. Each point represents the mean ± SD (range) of triplicate assays. Steady-state kinetic parameters were obtained with fitting to a standard Michaelis–Menten equation using the GraphPad Prism software (GraphPad). This is set in Prism as: Y = Vmax × X/(Km + X). The steady-state kinetic parameters are shown in Table 1.

**Figure 4 pharmaceutics-13-01429-f004:**
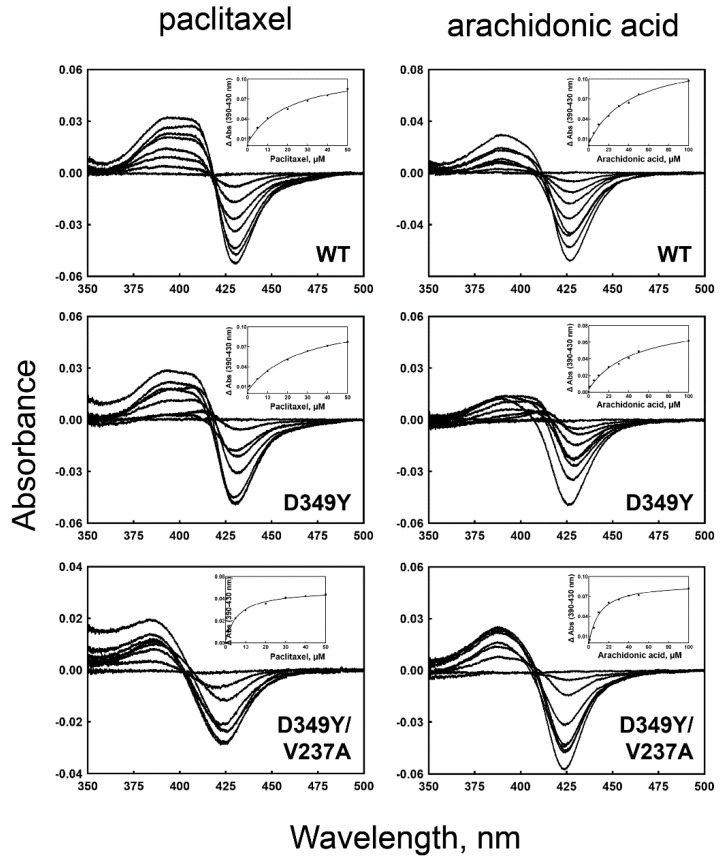
Fe^2+^-CO vs. Fe^2+^ binding spectra of purified wild-type and mutant CYP2C8 enzymes. The right panels show the binding titrations of paclitaxel, and the left panels show those of arachidonic acid. The insets show the plots of ΔA_390–430 nm_ versus the concentration of substrates. The binding parameters are shown in Table 2.

**Figure 5 pharmaceutics-13-01429-f005:**
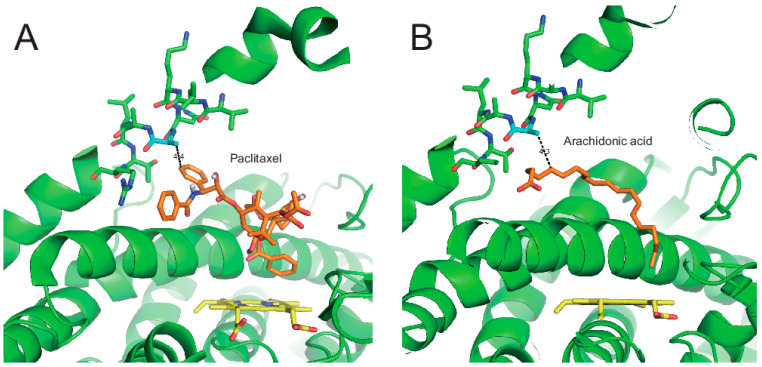
Molecular docking models of CYP2C8. The molecular docking models were constructed using the reported X-ray crystal structure of CYP2C8 (pdb ID 1PQ2). (**A**) Paclitaxel in the active site access channel of CYP2C8. The distance between the Ala237 residue of CYP2C8 and bupropion was measured at ~4.4 Å. (**B**) Arachidonic acid in CYP2C8. The distance between the Ala237 residue of CYP2C8 and arachidonic acid was measured at ~4.3 Å.

**Table 1 pharmaceutics-13-01429-t001:** Steady-state kinetic parameters of purified wild-type and mutant CYP2C8s.

**CYP2C8** **Mutants**	**6α-Hydroxylation of Paclitaxel**		
	***k*_cat_ (min^−1^)**	***K*_M_ (μM)**	***k*_cat_/*K*_M_**
WT	0.65 ± 0.03	7.4 ± 1.1	0.09 ± 0.01
D349Y	4.4 ± 0.2	11.1 ± 2.0	0.40 ± 0.04
D349Y/V237A	3.3 ± 0.2	15.0 ± 2.8	0.22 ± 0.04
	**ω-9 Epoxidation of Arachidonic Acid**		
	***k*_cat_ (min^−1^)**	***K*_M_ (μM)**	***k*_cat_/*K*_M_**
WT	0.055 ± 0.005	36.5 ± 8.5	0.0015 ± 0.0004
D349Y	1.6 ± 0.3	16.5 ± 10.2	0.10 ± 0.06
D349Y/V237A	5.6 ± 0.6	50.6 ± 12.8	0.11 ± 0.03
	**ω-6 Epoxidation of Arachidonic Acid**		
	***k*_cat_ (min^−1^)**	***K*_M_ (μM)**	***k*_cat_/*K*_M_**
WT	0.069 ± 0.013	19.4 ± 12.0	0.004 ± 0.002
D349Y	2.7 ± 0.5	17.2 ± 10.7	0.16 ± 0.10
D349Y/V237A	11.1 ± 1.0	45.1 ± 10.7	0.25 ± 0.06

Results are presented as means ± SD of triplicate assays.

**Table 2 pharmaceutics-13-01429-t002:** Estimated substrate affinities of wild-type and mutant CYP2C8s.

CYP2C8 Mutants	*K*_d_ (μM)	
	Paclitaxel	Arachidonic Acid
WT	20.3 ± 5.1	40.5 ± 6.0
D349Y	26.7 ± 5.6	44.5 ± 10.0
D349Y/V237A	5.7 ± 1.3	10.2 ± 1.5

## Data Availability

All data in this study have been included in this manuscript.

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
