# Peer review of "Structure-Functional Analysis of Human Cytochrome P450 2C8 Using Directed Evolution"

_pharmaceutics, 2021, doi:10.3390/pharmaceutics13091429_

Round 1
Reviewer 1 Report
In this manuscript, authors used directed evolution analysis to address the structure-functional complexity of CYP2C8. The work is meaning, however, it is has the following problems to solve before acceptance:
- Structure-functional analysis of human cytochrome P450 2C8 is important. However, the introduction of related works is relatively older and incomplete. I can not judge the significance of their work from the Section "Introduction". Suggest authors supply related the newest works.
- In the discussion, authors analyzed the experimental results. Suggest authors analyze the advantages and weakness of this work.
- In the conclusion, authors conclude their work. However, it is more appropriate to elaborate the further work in the future.
- Artificial intelligence techniques contribute to analyze structure-function relationship of human cytochrome P450 2C8. Suggest authors cooperate with researchers with artificial intelligence background to initially screen the relationship, and then analyze structure-function relationship of human cytochrome P450 2C8 by experiments. Artificial intelligence techniques have been widely applied to various relationship analyses, for example, PMID: 26731781, 32745502 used artificial intelligence methods to reposition FDA-approved drugs.
- English need to improve.
Author Response
1) Structure-functional analysis of human cytochrome P450 2C8 is important. However, the introduction of related works is relatively older and incomplete. I cannot judge the significance of their work from the Section "Introduction". Suggest authors supply related the newest works.
--> We added the previous studies to analyze the structure-functional analysis of CYP2C8 in Introduction section to improve the significance of this study. Thank you.
2) In the discussion, authors analyzed the experimental results. Suggest authors analyze the advantages and weakness of this work.
--> Successful directed evolution studies always require efficient high-throughput screening and finding a suitable screening method proved to be the most challenging aspect. Therefore, the luminescence is the selected approach to analysis of CYP2C8 activity. We described the limitation of this directed evolution and its advantage when analyzed with structural information in Discussion section. Thank you.
3) In the conclusion, authors conclude their work. However, it is more appropriate to elaborate the further work in the future. Artificial intelligence techniques contribute to analyze structure-function relationship of human cytochrome P450 2C8. Suggest authors cooperate with researchers with artificial intelligence background to initially screen the relationship, and then analyze structure-function relationship of human cytochrome P450 2C8 by experiments. Artificial intelligence techniques have been widely applied to various relationship analyses, for example, PMID: 26731781, 32745502 used artificial intelligence methods to reposition FDA-approved drugs.
--> We agreed that AI techniques will contribute to the directed evolution of P450s. We described many artificial intelligence techniques have developed, and they have been widely applied to various relationship analyses and used for the reposition FDA-approved drugs in Discussion section. We also discussed the cooperative work with artificial intelligence for future direction. Thank you.
4) English need to improve.
--> We edited all through the manuscript assisted by the Professional English Editing service (Editage). We attached the English editing certificate. Thank you.

Reviewer 2 Report
After revising the paper entitled “Structure-functional analysis of human cytochrome P450 2C8 using directed evolution”, I suggest the manuscript should undergo a major revision before publication in Pharmaceutics. The most notable areas for improvement are listed below:
Major comments:
Abstract
- The abstract should be a total of about 200 words maximum.
Materials and methods
- Collection of soluble fractions containing CYP2C8 wild type and mutants as well as the subsequent enzyme purification were referenced to Cho et al., 2019 and Yun et al., 2000. To facilitate the understanding of section 2.4. in materials and methods, more information should be provided in this regard. Similarly, some information should be provided about the obtention of rat NPR.
Results
- More information must be provided about how data in Figure 1B was obtained. Does each point represent the mean value obtained in triplicate assays? In addition, no error bars representing the SD values for wt and D349Y were provided. The SD for D349Y/V237A seems too high to consider the value as a statistically significant result.
- In Figure 2 caption, expression levels of wild type enzyme were ~500 nmol/L, while in the text those values were supposed to be exactly 500 nmol/L. This aspect must be clarified.
- When analyzing the steady-state kinetic values for arachidonic acid epoxidation, some of the kcat/KM values differ from values calculated directly from data in Table 1. This aspect must be thoroughly addressed.
- It seems obvious that the catalytic efficiencies of mutant enzymes were significantly higher than values observed with the wild type enzyme. However, some SD values for the epoxidation of arachidonic acid seem too high to consider these values as meaningful result.
- In Figure 3, the mathematical model used to connect individual points in the chart should be provided.
Discussion and conclusion
- The authors make some interesting and plausible assumptions based on the previously published structural data. Authors claim that “the modulation of architectures in the substrate access channel and reduction partner binding domain dramatically influences the catalytic activity”, but they do not provide novel data in this regard. Thus, crystallographic experiments (or MD simulations) are imperative to support authors’ statements.
Minor comments
- “E. coli” should be written in italics.
- In all cases, nmol P450/liter culture should be written as nmol/L.
- “Km” should be written as KM
- In Table 1 “6α-Hydroxylation of paclitaxel” should not be written in bold.
- In Table 2 units for Kd values should be written in parentheses: (µM)
- The second sentence on Figure 4 caption must be rewritten.
Author Response
[Reviewer #2]
1) Abstract: The abstract should be a total of about 200 words maximum.
--> We revised the abstract with 200 words. Thank you.
2) Materials and methods: Collection of soluble fractions containing CYP2C8 wild type and mutants as well as the subsequent enzyme purification were referenced to Cho et al., 2019 and Yun et al., 2000. To facilitate the understanding of section 2.4. in materials and methods, more information should be provided in this regard. Similarly, some information should be provided about the obtention of rat NPR.
--> We added detailed procedure to purified P450 and NPR in Materials and methods section. Thank you.
3) Results: More information must be provided about how data in Figure 1B was obtained. Does each point represent the mean value obtained in triplicate assays? In addition, no error bars representing the SD values for wt and D349Y were provided. The SD for D349Y/V237A seems too high to consider the value as a statistically significant result.
--> After initial screening at each round, the luminescent activity produced by the selected mutant clones was remeasured in triplicate assays. We described this in the Figure legend.
There are absolutely error bars in wt and D349Y but they are hard to be seen in the same scaled figure the because they are much smaller than that of scale. When two screening rounds are separated, the error bars are clearly seen as following.
And the SD for the activities of D349Y/V237A was less than 20% but we believe they are reasonable because these results were directedly from the whole cell of E. coli. The luminescent activities of purified mutants with normalized P450 content were presented in Fig 3A. Thank you.
4) In Figure 2 caption, expression levels of wild type enzyme were ~500 nmol/L, while in the text those values were supposed to be exactly 500 nmol/L. This aspect must be clarified.
--> Usually, the measurement of P450 expression level in E. coli whole cell is not so exactly accurate. Therefore, we described ~500 nmol/L in the Figure legend. We corrected it in the main text in Result section. Thank you.
5) When analyzing the steady-state kinetic values for arachidonic acid epoxidation, some of the kcat/KM values differ from values calculated directly from data in Table 1. This aspect must be thoroughly addressed.
--> There were calculation errors to obtain kcat/KM values in Table 1. We recalculated and verified the values. The corrected values were revised in the manuscript. Thank you.
6) It seems obvious that the catalytic efficiencies of mutant enzymes were significantly higher than values observed with the wild type enzyme. However, some SD values for the epoxidation of arachidonic acid seem too high to consider these values as meaningful result.
--> The high SD values of mutant in the catalytic efficiencies are due to the high SDs in Km values. SD values in steady-state kinetics analysis are the deviation from fitting not from the individual data points (See the Figure 3C and 3D). If we use the data up to 100 uM, the SD values decrease but the fittings do not saturate.
7) In Figure 3, the mathematical model used to connect individual points in the chart should be provided.
--> We performed steady-state kinetic kinetics analysis with a standard Michaelis-Menten equation using GraphPad Prism software. This is set in Prism as: Y=Vmax*X/(Km+X). We described this in Figure 3 legend. Thank you.
8) Discussion and conclusion: The authors make some interesting and plausible assumptions based on the previously published structural data. Authors claim that “the modulation of architectures in the substrate access channel and reduction partner binding domain dramatically influences the catalytic activity”, but they do not provide novel data in this regard. Thus, crystallographic experiments (or MD simulations) are imperative to support authors’ statements.
--> As suggested by the reviewer, we added this statement in Discussion. However, the experiment including crystallography is beyond the scope of this study. Thank you.
9) Minor comments:
“E. coli” should be written in italics.
In all cases, nmol P450/liter culture should be written as nmol/L.
“Km” should be written as KM
In Table 1 “6α-Hydroxylation of paclitaxel” should not be written in bold.
In Table 2 units for Kd values should be written in parentheses: (µM)
The second sentence on Figure 4 caption must be rewritten.
--> All corrected. Thank you.

Round 2
Reviewer 2 Report
Despite the authors have changed some details of the manuscript there is a major issue that should be revised. First of all, the quality of figure 5 is non-acceptable for publishing (according to my opinion). Additionally, the discussion of the structural data is not supported by the sole presence of this figure. As previously suggested, authors should simulate these mutations and perform a molecular docking (at least) of the complex wild-type enzyme-substrate, and another for the complex mutated enzyme-substrate to validate their hypothesis.
Author Response
Response: As the reviewer suggested, we constructed the molecular docking models to validate the role of Ala237 residue to modulate the substrate binding through the active site access channel. In the docking models, paclitaxel and arachidonic acid interacted with Ala237 residue with close distances of 4.4 Å and 4.3 Å, respectively. These docking models implies that the V237A mutation can modulate substrates binding in the active site access channel. We made a new Figure 5 showing the docking model analysis. However, Because the D349Y mutation in helix K is outside of the protein and the X-ray crystal structure of reductase complex is not currently available, our postulation of D349Y mutation was obtained from the previous indirect supporting studies (Estrada, 2013; Kim, 2021) and the location of Asp349 in the structure of CYP2C8, which is now presented as Supporting data (Figure S1).
Thank you for the excellent advice to improve the quality of our study.

Round 3
Reviewer 2 Report
The manuscript is suitable to be published